# Blockade of IKK signaling induces RIPK1-independent apoptosis in human macrophages

Neha M. Nataraj[1,2,3], Reyna Garcia Sillas[2], Beatrice I. Herrmann[2], Sunny Shin[1,3]*, Igor E. Brodsky [1,2]*

1 Institute for Immunology & Immune Health, University of Pennsylvania Perelman School of Medicine, Philadelphia, Pennsylvania, United States of America, 2 Department of Pathobiology, University of Pennsylvania School of Veterinary Medicine, Philadelphia, Pennsylvania, United States of America, 3 Department of Microbiology, University of Pennsylvania Perelman School of Medicine, Philadelphia, Pennsylvania, United States of America

* sunshin@pennmedicine.upenn.edu (SS); ibrodsky@vet.upenn.edu (IEB)

## Abstract

Regulated cell death in response to microbial infection plays an important role in immune defense and is triggered by pathogen disruption of essential cellular pathways. Gram-negative bacterial pathogens in the *Yersinia* genus disrupt NF-κB signaling via translocated effectors injected by a type III secretion system, thereby preventing induction of cytokine production and antimicrobial defense. In murine models of infection, *Yersinia* blockade of NF-κB signaling triggers cell-extrinsic apoptosis through Receptor Interacting Serine-Threonine Protein Kinase 1 (RIPK1) and caspase-8, which is required for bacterial clearance and host survival. Unexpectedly, we find that human macrophages undergo apoptosis independently of RIPK1 in response to *Yersinia* or chemical blockade of IKKβ. Instead, IKK blockade led to decreased cFLIP expression, and overexpression of cFLIP contributed to protection from IKK blockade-induced apoptosis in human macrophages. We found that IKK blockade also induces RIPK1 kinase activity-independent apoptosis in human T cells and human pancreatic cells. Altogether, our data indicate that, in contrast to murine cells, blockade of IKK activity in human cells triggers a distinct apoptosis pathway that is independent of RIPK1 kinase activity. These findings have implications for the contribution of RIPK1 to cell death in human cells and the efficacy of RIPK1 inhibition in human diseases.

## Author summary

Programmed cell death is critical for organismal homeostasis and for host defense against microbial infection. Gram-negative bacteria of the genus *Yersinia* cause diseases ranging from plague (*Y. pestis*) to severe gastroenteritis (*Y. pseudotuberculosis* and *Y. enterocolitica*). Studies in murine models have demonstrated that all pathogenic *Yersinia* disrupt pro-inflammatory cell signaling, which triggers cell death in murine macrophages. This cell death is mediated by the kinase RIPK1, and RIPK1-mediated cell death is essential for host defense and survival. Although murine studies have provided insight into

**Data Availability Statement:** Raw data is available on Dryad. https://doi.org/10.5061/dryad.hmgqnk9rz.

**Funding:** This work was supported by NIH grants AI139102A1 (I.E.B.), AI125924 (I.E.B.), AI118861 (S.S.), AI123243 (S.S.), the Burroughs Wellcome

Fund Investigator in the Pathogenesis of Infectious Disease awards (both I.E.B. and S.S.), T32 AI141393 (N.M.N.), the American Heart Association Predoctoral Fellowship (N.M.N.), and NIH NRSA training grant AI161319-03 (B.I.H.). The funders had no role in study design, data collection and analysis, decision to publish, or preparation of the manuscript.

**Competing interests:** The authors have declared that no competing interests exist.

mechanisms that regulate host defense during *Yersinia* infection, there are important differences between human and murine immune systems regarding expression and presence of key proteins. Thus, how human cells respond to *Yersinia* infection remains poorly understood. Here, we report that, in contrast to murine systems, *Yersinia* infection or chemical blockade of immune signaling in human cells induces a distinct apoptotic cell death that is entirely independent of RIPK1 kinase activity. RIPK1 is implicated in a wide range of systemic disorders and is currently being targeted in clinical trials. Our study provides insight into human-specific cell death signaling and host defense mechanisms, which has implications for understanding human immune responses to infection and therapeutic approaches for treating human diseases.

## Introduction

Regulated cell death in response to microbial infection is critical for immune defense. Innate immune cells utilize pattern recognition receptors to detect pathogen-associated molecular patterns and elicit an inflammatory response [1,2]. To evade this detection, many microbial pathogens employ mechanisms to suppress immune signaling [3,4]. In response, the host has evolved compensatory mechanisms to induce cell death and inflammation in response to pathogen-mediated disruption of TNFR superfamily [5,6] and TLR signaling [6,7].

Stimulation of TNFR or TLR4 induces recruitment of Receptor Interacting Serine-Threonine Protein Kinase 1 (RIPK1) to the receptor proximal signaling complex. RIPK1 serves as a Complex 1 scaffold for downstream signaling proteins, including TAK1 and IKKα/β, which phosphorylate RIPK1 to maintain its localization [8–15]. The kinases in Complex I initiate NF-κB and MAPK signaling, resulting in inflammatory cytokine production [16]. However, pathogen-mediated or pharmacological blockade of key Complex I proteins, particularly TAK1 and IKKα/β, destabilizes the complex. Released RIPK1 then recruits FADD and caspase-8 to form Complex II, mediating rapid caspase 8-dependent cell death [8,10,17].

The Gram-negative bacterial genus *Yersinia* is a natural pathogen of both rodents and humans and is responsible for human diseases ranging from gastroenteritis to plague [18,19]. *Yersinia* utilizes a type III secretion system (T3SS) to inject virulence factors known as *Yersinia* outer proteins (Yops) into the host cell cytosol [20,21]. One of these Yops, known as YopP in *Y. enterocolitica* and YopJ in *Y. pseudotuberculosis*, potently blocks IKKβ and TAK1 in murine macrophages, resulting in RIPK1- and caspase-8-dependent cell death [14,22–35]. Importantly, RIPK1 kinase activity is critical for caspase-8-dependent cell death, restriction of bacterial loads, and host survival during *Yersinia* infection in mice [12,33].

While murine models have been vital to elucidate mechanisms of cell death and inflammation during *Yersinia* infection, there exist notable differences between human and murine immune systems [36–39], indicating that mice do not fully recapitulate human responses to infection. Notably, mice lacking RIPK1 experience acute perinatal lethality, succumbing to systemic inflammation and aberrant cell death [40], whereas humans with biallelic RIPK1 deficiency are viable, although they experience immunodeficiencies and autoinflammatory disease [41,42]. Clinical studies have implicated human RIPK1 in a wide range of systemic disorders and pathologies [41–46], leading to substantial interest in developing RIPK1 therapeutics, particularly for treating inflammatory diseases and cancer [46–48]. However, whether human and murine cells similarly undergo RIPK1-dependent cell death in response to blockade of IKK signaling remains poorly understood.

Here, we demonstrate that in human macrophages, cell death induced by *Yersinia* or chemical IKK blockade is independent of RIPK1, indicating that regulation of apoptosis in human macrophages is distinct from mouse macrophages. Instead, our data suggest that cell death is caused by IKK blockade-mediated downregulation of cFLIP. cFLIP overexpression partially protected human macrophages from IKK blockade-induced cell death. Moreover, we found that RIPK1 activity is also dispensable for apoptosis of human Jurkat T cells and pancreatic cells following IKK blockade. Altogether, our data demonstrate that IKK blockade triggers apoptosis independently of RIPK1 activity in multiple human cell types. Our findings suggest that investigation of compensatory cell death pathways is warranted in innate immune responses to bacterial infection and in the setting of therapeutics targeting RIPK1 for human disease.

## Results

### *Yersinia* YopP blockade of IKK signaling induces RIPK1 activity-independent apoptosis in human macrophages

*Yersinia* YopJ blockade of IKK signaling induces rapid RIPK1- and caspase-8-dependent cell death in murine bone marrow-derived macrophages (BMDMs) [12,23,31,33–35] (**S1A Fig**). However, primary human monocyte-derived macrophages (hMDMs) infected with *Y. pseudotuberculosis* (*Yptb*) did not undergo detectable cytotoxicity (**S1B Fig**), consistent with recent findings [34]. The related species *Y. enterocolitica* (*Ye*) expresses a YopJ homolog, termed YopP, that is required for *Ye* to block TNF production in hMDMs (**Fig 1A**) [22,49]. In contrast to *Yptb*, *Ye* induced robust YopP-dependent cell death in both hMDMs and BMDMs (**S1A–S1C Fig**), as previously reported [50]. Interestingly, a Δ*yopJ Yptb* strain expressing YopP was not sufficient to recapitulate the cell death induced by WT *Ye* (**S1C Fig**). Unexpectedly, Necrostatin-1 (Nec-1), a small molecule inhibitor of RIPK1 kinase activity, did not block *Ye*-induced death of hMDMs (**Fig 1B and 1C**), in contrast to its inhibitory effect on cell death in *Yersinia*-infected murine BMDMs (**S1A Fig**) [31,34].

Immunoblot analysis of hMDMs demonstrated that YopP-dependent caspase-8 processing into its active form was not inhibited by Nec-1 (**Fig 1D**), in contrast to BMDMs. *Ye* also induced YopP-dependent cleavage of the cell-intrinsic initiator caspase, caspase-9, which was not blocked with Nec-1 (**S1D Fig**), consistent with observations that *Yersinia* induces caspase-8-dependent cleavage of BID, which can activate the mitochondrial apoptosis pathway [25]. The downstream executioner caspase-3 was processed into the active subunits p17/p19 in a RIPK1 activity-independent manner (**Fig 1E**). Human THP-1 macrophages also exhibited robust YopP-dependent, RIPK1 activity-independent caspase-3/7 cleavage and activation (**Fig 1F and 1G**). Overall, these data demonstrate that *Ye* induces YopP-dependent, RIPK1 activity-independent apoptosis in human macrophages.

### RIPK1 activity is dispensable for IKK blockade-induced apoptosis of human macrophages

In murine BMDMs, TNF or TLR stimulation together with pharmacological TAK1 or IKK blockade induces RIPK1- and caspase-8-dependent cell death [12,34], similar to *Yersinia* infection. Consistently, hMDMs (**Fig 2A and 2B**) and THP-1 macrophages (**Fig 2C**) treated with LPS and an IKK inhibitor (LPS+IKKi) underwent cell death. However, as with *Yersinia* infection (**Fig 1**), this cell death was not blocked by Nec-1, indicating that RIPK1 activity was not required. hMDMs treated with LPS and a TAK1 inhibitor (LPS+TAK1i) also underwent cell death that was not significantly blocked by Nec-1 (**S2A Fig**). In murine cells, Complex I

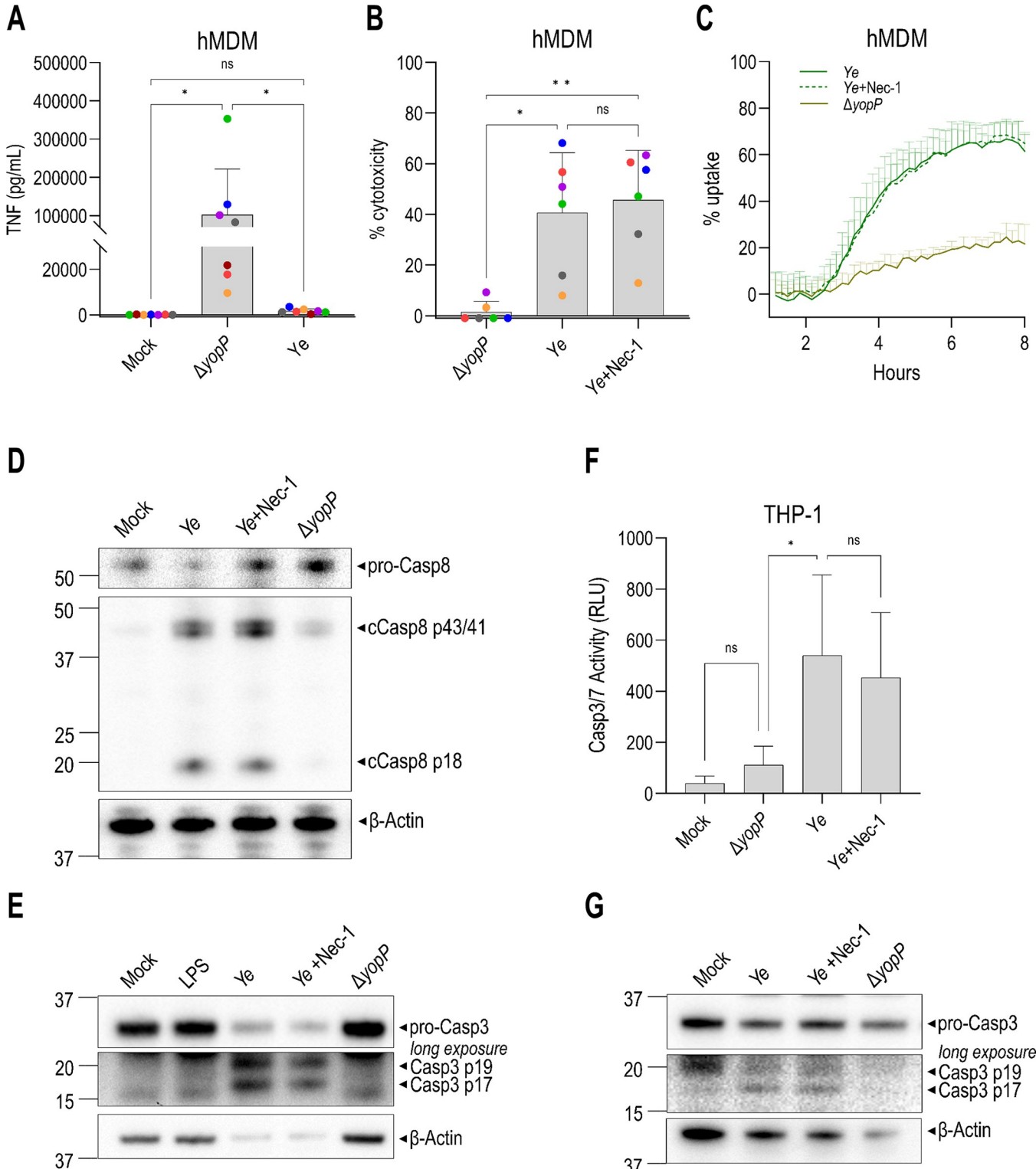

**Fig 1. RIPK1 activity is dispensable for *Yersinia*-induced extrinsic apoptosis in human macrophages.** Cells were pre-treated with Nec-1 or a media control for 1 h and then mock-infected or infected with WT *Y. enterocolitica* (*Ye*) or Δ*yopP Ye*. **(A–B)** Each data point represents the mean of triplicate wells for each of 6–7 different human donor hMDMs. **(A)** TNF levels in the supernatant of hMDMs were measured by ELISA 18–19 h after infection. N = 7. **(B)** Cytotoxicity was measured by lactate dehydrogenase (LDH) release from hMDMs at 18–19 h after infection and normalized to untreated cells. **(C)** Cytotoxicity in hMDMs was measured by PI uptake in triplicate wells over 8 h. Representative of 3 independent experiments with different human donors. **(D)** Immunoblot analysis was performed on hMDM lysates 5 h after infection for full-length and cleaved caspase-8, and β-actin. Representative of 3 independent experiments with

different human donors. **(E)** Immunoblot analysis was performed on hMDM lysates 18 h after infection for caspase-3 and β-actin. Representative of 3 independent experiments with different human donors. **(F)** Caspase-3/7 activity in THP-1 macrophages quantified 20–26 h after infection. N = 5. **(G)** Immunoblot analysis of THP-1 lysates 24 h after infection for caspase-3 and β-actin. Representative of 2 independent experiments. ns, not significant, *p < 0.05, **p < 0.01, ***p < 0.001, ****p < 0.0001 by Tukey's multiple comparisons test. Graphs depict mean + SD.

blockade induces RIPK1 autophosphorylation, which promotes recruitment of RIPK1 into a stable Complex II to robustly activate caspase-8 [10,51,52]. However, we did not detect RIPK1 phosphorylation following LPS+IKKi treatment in hMDMs (**Figs 2D and S2B**), but we did detect RIPK1 cleavage to its inactive form, likely by caspase-8 to prevent necroptosis as characterized previously [45,53,54]. Furthermore, we also did not observe potent RIPK1 phosphorylation in hMDMs following co-treatment with LPS, IKKi, and an inhibitor of p38 MAPK (LPS+IKKi+p38i), nor with LPS+TAK1i treatment (**S2B Fig**).

We also treated cells with LPS and the pan-caspase inhibitor ZVAD, which engages RIPK1-dependent necroptosis by blocking caspase-8 activity [55]. Importantly, we observed robust cell death that was effectively suppressed by Nec-1 (**Fig 2A–2C**). Furthermore, there was early and robust RIPK1 phosphorylation following LPS+ZVAD treatment (**Figs 2D and S2B**), which was effectively suppressed by Nec-1. Consistent with other studies [56], LPS+ZVAD-induced necroptosis of human macrophages was blocked by the RIPK3 inhibitor GSK'872 (**S2C Fig**). In total, our data indicate that IKK blockade-induced death in human macrophages occurs independently of RIPK1 activity, whereas necroptosis in human macrophages is dependent on RIPK1 activity.

As with *Ye* infection, LPS+IKKi treatment induced processing of both caspase-3 (**Fig 2D and 2E**) and caspase-8 (**Fig 2F**) into their active forms, which was not blocked by Nec-1. Moreover, LPS+IKKi-induced cell death was significantly reduced upon co-treatment with GSK'872 and the caspase-8 inhibitor IETD in hMDMs, or with GSK'872 and the pan-caspase inhibitor QVD in THP-1 macrophages (**S2D Fig**), consistent with the interpretation that LPS+IKKi induces caspase-8-dependent cell death. LPS+IKKi also induced RIPK1 activity-independent caspase-9 cleavage (**S2E Fig**), as observed with *Ye* infection (**S1D Fig**). Furthermore, we observed that caspase-8 and caspase-9 processing into their active forms both occurred rapidly by 2 hours following LPS+IKKi treatment or *Ye* infection (**Fig 2G**), indicating that both the extrinsic and intrinsic apoptosis pathways are activated concurrently.

To address the potential contribution of the mitochondrial apoptosis pathway to IKK blockade-induced apoptosis in human macrophages, we used siRNA to silence *BID* expression in THP-1 macrophages. Despite substantial reduction in BID protein levels (**S2F Fig**), we found no significant reduction in cytotoxicity following LPS+IKKi treatment (**S2G Fig**). These data indicate that although the mitochondrial apoptosis pathway is activated following IKK blockade, it is not required to execute cell death, presumably because the cell-extrinsic pathway is sufficient.

In murine macrophages, the *Yersinia*-induced RIPK1/caspase-8 Complex II contributes to pyroptosis by activating the pore-forming protein Gasdermin D (GSDMD) and promoting IL-18 release [31,34,35,57]. However, neither LPS+IKKi treatment nor *Yersinia* infection induced IL-18 release in THP-1 macrophages, suggesting that human macrophages do not undergo pyroptosis or inflammasome activation in response to IKK blockade (**S2H Fig**). As expected, LPS+Nigericin treatment, which activates the NLRP3 inflammasome, induced IL-18 release in THP-1 macrophages, which was abrogated by the NLRP3 inhibitor MCC950 (**S2H Fig**).

To further characterize the regulation of IKK blockade-induced cell death, we used CRISPR/Cas9 to generate three independent *FADD*[-/-] single-cell clonal cell lines (**S2I and S2J Fig**). Following LPS stimulation, *FADD*[-/-] THP-1 macrophages underwent cell death, and

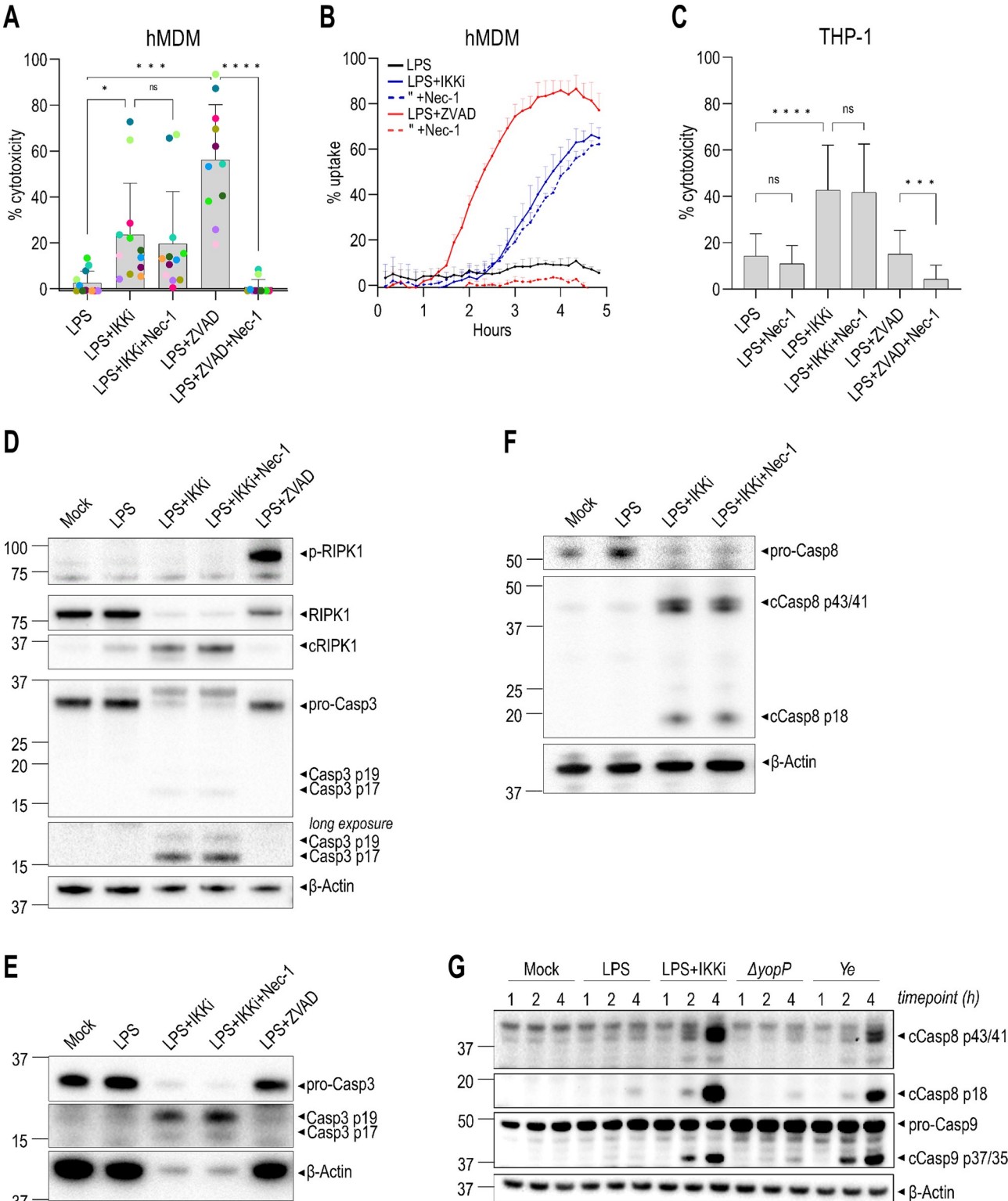

**Fig 2. RIPK1 activity is dispensable for IKK blockade-induced extrinsic apoptosis in human macrophages.** Cells were pre-treated with IKKi, Nec-1, and/or ZVAD and then stimulated with LPS. **(A)** Cytotoxicity in hMDMs was measured by LDH release 5–7 h post-stimulation. Each data point represents the mean of triplicate wells for 11–12 different human donors. **(B)** Cytotoxicity in hMDMs was measured by PI uptake in triplicate wells over 5 h. Representative of 3 independent experiments. **(C)** Cytotoxicity in THP-1 macrophages was measured by LDH release 17–24 h post-stimulation. N = 22–35. **(D–G)** Immunoblot analysis was performed on lysates as indicated: **(D)** hMDM lysates 6 h after stimulation for phospho-RIPK1 (S166),

RIPK1, caspase-3, and β-actin, representative of 3–4 independent experiments, (**E**) THP-1 lysates 24 h after stimulation for caspase-3 and β-actin, representative of 2 independent experiments, (**F**) hMDM lysates 5 h after stimulation for full-length and cleaved caspase-8, and β-actin, representative of 3 independent experiments, and (**G**) hMDM lysates at various time points for cleaved caspase-8, caspase-9, and β-actin, representative of 2–3 independent experiments. ns, not significant, *p < 0.05, **p < 0.01, ***p < 0.001, ****p < 0.0001 by Tukey's multiple comparisons test. Graphs depict mean + SD.

upon treatment with Nec-1 to block RIPK1 activity or with GSK'872 to block RIPK3 activity, there was significantly less cell death (**S2K Fig**), indicating that *FADD*[-/-] cells are predisposed to undergo necroptosis in response to LPS. Thus, upon LPS stimulation of human macrophages, FADD appears to protect cells from undergoing necroptosis, likely due to the role of FADD in stabilizing caspase-8:cFLIP heterodimers to promote their pro-survival and anti-necroptosis functions [58–60]. Importantly, *FADD*[-/-] cells also undergo significantly less cell death following LPS+IKKi treatment (**S2L Fig**) and significantly less caspase-3/7 activation following *Yersinia* infection (**S2M Fig**). Thus, FADD appears to be largely required for this cell-extrinsic apoptosis pathway in human macrophages. Collectively, our data indicate that blockade of IKK signaling in human macrophages induces caspase-8 activation and FADD-mediated apoptosis independently of RIPK1 activity.

## RIPK1 is dispensable for IKK blockade-induced cell-extrinsic apoptosis in human macrophages

In murine cells, RIPK1 can contribute to cell-extrinsic apoptosis via both kinase-dependent and independent functions [8–12,15,51,61,62]. Blockade of Complex I proteins at Checkpoint 1 occurs upstream of NF-κB-dependent gene regulation and triggers rapid RIPK1 activity-dependent cell death [7–12,17,51,63]. Checkpoint 2 occurs at the level of transcriptional/translational regulation of pro-survival genes such as cFLIP and A20 [7,10,15,17,51,61]. Downregulation of cFLIP leads to the loss of caspase-8-cFLIP heterodimers that normally inhibit cell death, inducing RIPK1 activity-independent cell death [6,10,17,51,62]. As our data suggested that IKK blockade-induced apoptosis in human macrophages is independent of RIPK1 kinase activity, we next directly tested the requirement for RIPK1 in IKK blockade-induced cell death. We used CRISPR/Cas9 to generate two independent, sequence-validated *RIPK1*[-/-] THP-1 single-cell clonal cell lines (**S3A–S3D Fig**). *RIPK1*[-/-] THP-1 macrophages released significantly lower levels of TNF upon LPS stimulation (**Fig 3A**), consistent with the reported role for RIPK1 in TLR4 signaling [64]. *RIPK1*[-/-] THP-1 macrophages were not sensitized to LPS-induced cytotoxicity (**Fig 3B**), similar to previous findings [42], but in contrast to other findings with human iPSC-derived macrophages stimulated with LPS [65] and *RIPK1*[-/-] Jurkat T cells stimulated with TNF [8,17,66].

Following LPS+IKKi treatment or *Ye* infection, WT and *RIPK1*[-/-] THP-1 macrophages exhibited comparable levels of cell death (**Fig 3B**) and caspase-3/7 activation (**Figs 3C, 3D and S3E**). Consistent with the critical role of RIPK1 in necroptosis, *RIPK1*[-/-] THP-1 macrophages were protected from cytotoxicity in response to LPS+ZVAD (**Fig 3B**). Altogether, these findings demonstrate that human macrophages do not require RIPK1 for IKK blockade-induced apoptosis.

In some contexts, the protein TNFR1-associated death domain (TRADD) can serve as a redundant scaffolding protein to promote Complex II assembly and mediate apoptosis independently of RIPK1. Like RIPK1, TRADD is a multifunctional protein that regulates both pro-inflammatory gene expression and cell death downstream of cell surface receptor ligation by recruiting distinct signaling complexes [13,59,67–71]. Notably, both hMDMs and THP-1 macrophages express high levels of TRADD, in contrast to murine BMDMs (**S3F Fig**).

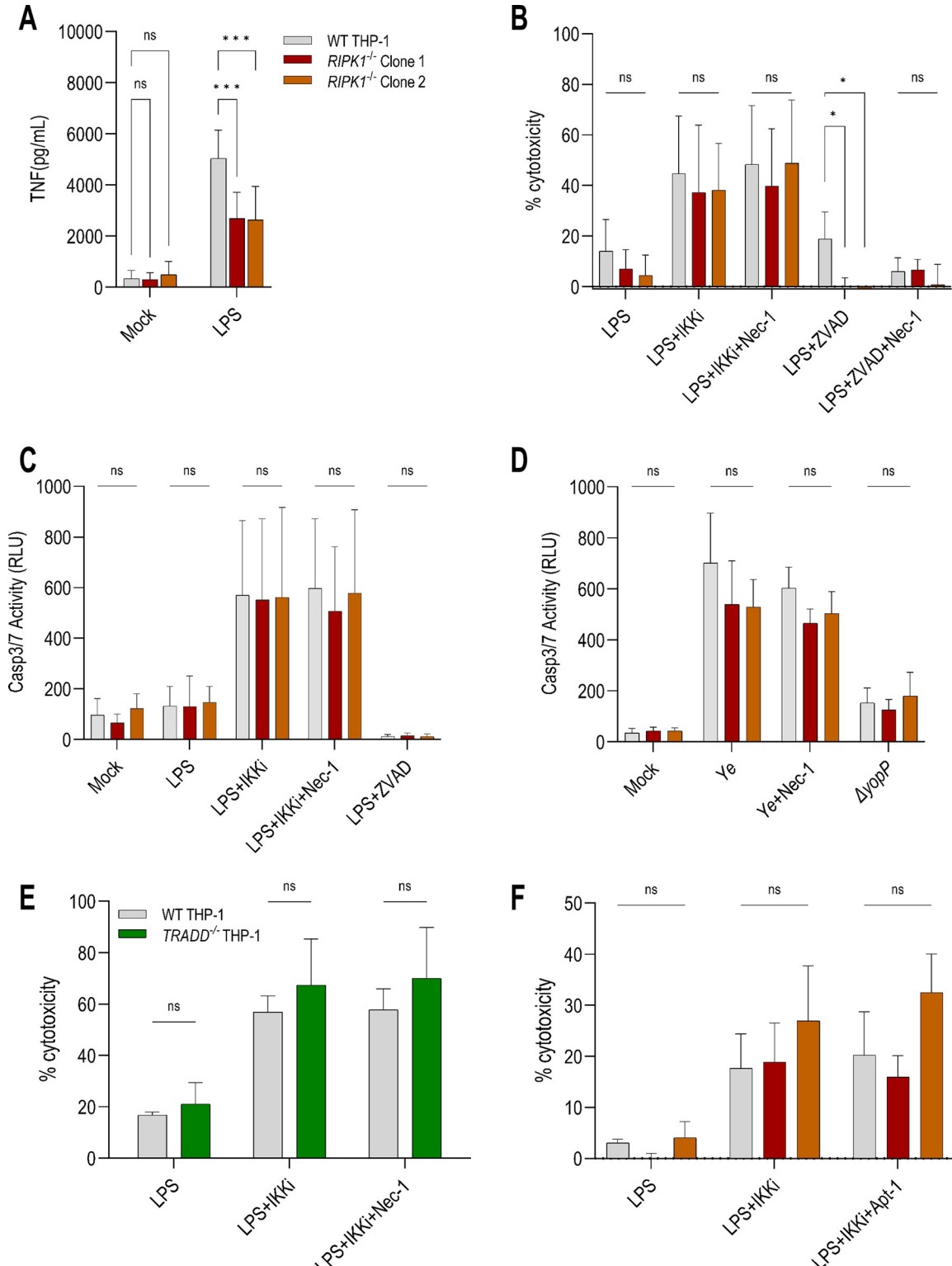

**Fig 3. Human *RIPK1*<sup>-/-</sup> macrophages undergo cell-extrinsic apoptosis following IKK blockade and *Yersinia* infection. (A–D)** WT and *RIPK1*<sup>-/-</sup> THP-1 macrophages were pre-treated with inhibitors and then stimulated with LPS or infected with *Ye* (WT or Δ*yopP*). **(A)** WT and *RIPK1*<sup>-/-</sup> THP-1 macrophages were stimulated with LPS for 18–24 h. TNF levels were measured in the supernatant by ELISA. **(B)** Cytotoxicity was measured by LDH release 18–24 h post-stimulation. N = 6–14. **(C,D)** Caspase-3/7 activity was detected and quantified by Caspase-Glo 3/7 22–24 h after **(C)** stimulation, N = 3–4 or **(D)** infection, N = 3. **(E,F)** WT, *RIPK1*<sup>-/-</sup>, and *TRADD*<sup>-/-</sup> THP-

1 macrophages were pre-treated with IKKi, Nec-1, and/or Apt-1 and then stimulated with LPS. Cytotoxicity was measured by LDH release (E) 20–24 h after stimulation, N = 3 and (F) 13–14 h after stimulation, N = 3. ns, not significant, *p < 0.05, **p < 0.01, ***p < 0.001 by (A–D,F) Tukey's or (E) Šídák's multiple comparisons test. Graphs depict mean + SD.

To directly test the requirement for TRADD, we generated *TRADD*⁻/⁻ THP-1 macrophages using CRISPR/Cas9 (**S3G Fig**) and utilized the bulk population, which demonstrated efficient TRADD depletion at the protein level (**S3H Fig**). LPS+IKKi treatment induced comparable levels of cell death between WT and *TRADD*⁻/⁻ THP-1 macrophages (**Fig 3E**), indicating that TRADD is not required to mediate cell death following IKK blockade. To account for the possibility of functional redundancy between TRADD and RIPK1, we treated *TRADD*⁻/⁻ THP-1 macrophages with Nec-1, which failed to restrict cytotoxicity (**Fig 3E**). Furthermore, we treated cells with the small-molecule compound Apostatin-1 (Apt-1), which prevents TRADD from assembling Complex II [71], and treatment with this too failed to block LPS+IKKi-induced cell death in both WT and *RIPK1*⁻/⁻ THP-1 macrophages (**Fig 3F**). In total, these data suggest that human macrophages do not require TRADD nor RIPK1 to regulate IKK blockade-induced apoptosis.

## cFLIP partially protects human macrophages from IKK blockade-induced apoptosis

Since RIPK1 is dispensable for IKK blockade-induced apoptosis in human macrophages, we hypothesized that inhibition of IKK reduces expression of pro-survival genes such as *CFLAR* (encoding cFLIP), thus relieving a brake on cell-extrinsic apoptosis [17,55,57,58,72]. Indeed, we observed rapid downregulation of cFLIP expression following LPS+IKKi treatment (**Fig 4A**). Consistently, we also observed rapid cFLIP downregulation following LPS+Cycloheximide (Chx) treatment (**S4A Fig**). We generated two independent, sequence-validated *CFLAR*⁻/⁻ THP-1 single-cell clonal cell lines (**S4B–S4D Fig**). Notably, *CFLAR*⁻/⁻ THP-1 macrophages were significantly more susceptible than WT cells to caspase-3/7 activation and cell death following LPS or TNF stimulation alone (**Fig 4B and 4C**), consistent with the established protective role of cFLIP in preventing cell-extrinsic apoptosis [17,55,57,58,72]. Cytotoxicity of LPS+IKKi-treated WT cells was comparable to that of *CFLAR*⁻/⁻ cells, suggesting that IKKi-induced downregulation of cFLIP sensitizes THP-1 macrophages to death similarly to cells genetically lacking cFLIP (**Fig 4B**). Furthermore, LPS+ZVAD treatment largely protected *CFLAR*⁻/⁻ cells from LPS-induced cell death (**Fig 4B**), consistent with our finding that *CFLAR*⁻/⁻ cells are predisposed to undergo caspase-3/7 activation and apoptosis in response to LPS (**Fig 4C**).

We next tested whether cFLIP overexpression would protect human macrophages from IKK blockade-induced apoptosis by generating THP-1 cells stably overexpressing either cFLIP (plenti-*CFLAR*) or *mCherry* as a negative control (plenti-*mcherry*) (**S4E Fig**). Indeed, THP-1 macrophages overexpressing cFLIP were significantly protected from cell death following *Yersinia* infection or LPS+IKKi treatment (**Fig 4D–4G**). Furthermore, we observed reduced caspase-8 and caspase-9 cleavage in THP-1 macrophages overexpressing cFLIP (**S4F Fig**). However, protection from cell death was incomplete, indicating that although cFLIP is necessary, its overexpression is not sufficient to fully protect cells from IKK blockade-induced cell death. Altogether, these data indicate that cFLIP overexpression partially protects against IKK blockade-induced apoptosis in human macrophages.

## RIPK1 kinase activity is dispensable for IKK blockade-induced cell death in multiple human cell types

We next considered whether the lack of a role for RIPK1 activity in regulating IKK blockade-induced apoptosis was specific to human macrophages or a feature shared by other human

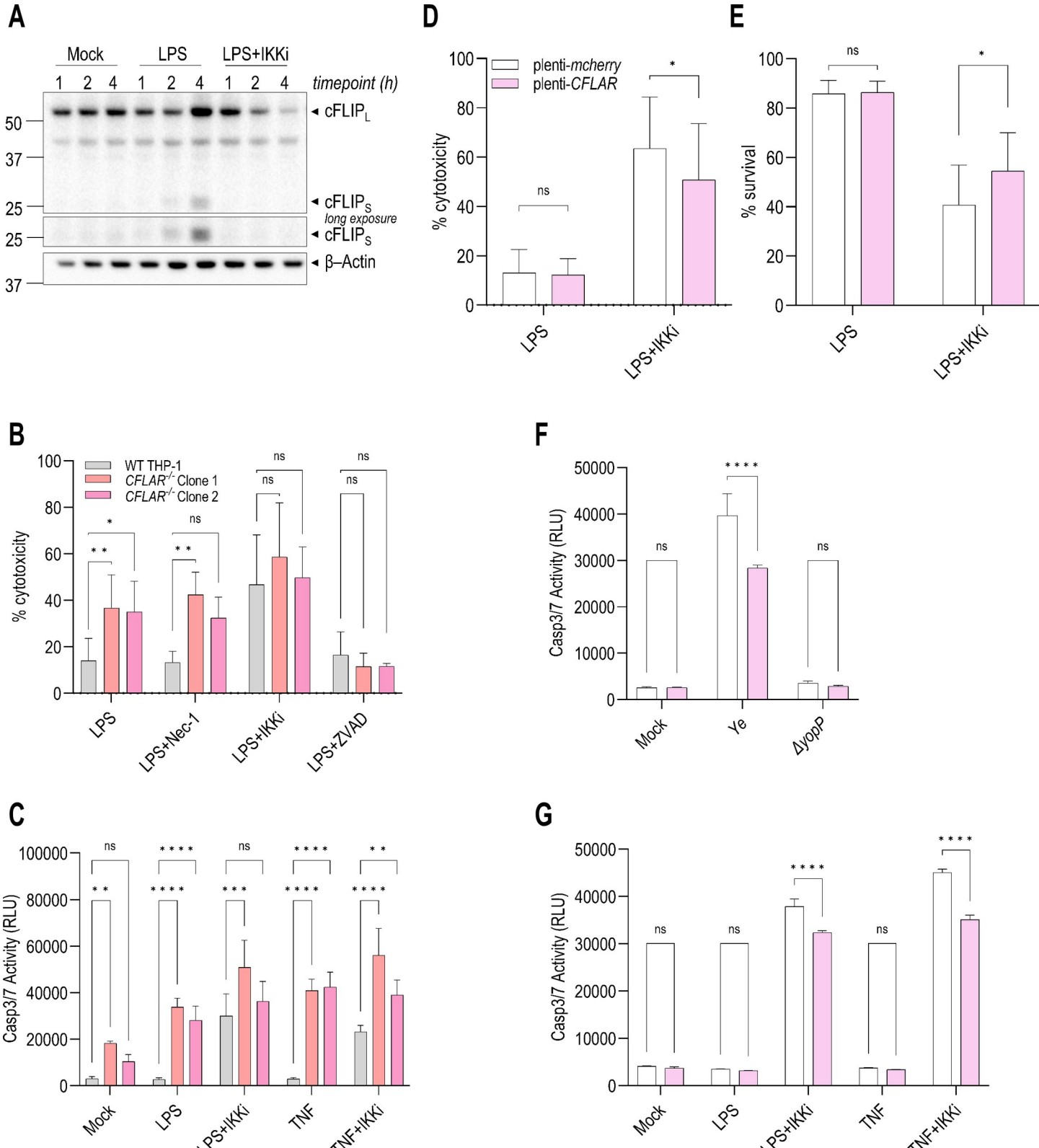

**Fig 4. cFLIP partially regulates extrinsic apoptosis in human macrophages following IKK blockade.** (A) hMDMs were pre-treated with IKKi and then stimulated with LPS. Immunoblot analysis was performed on lysates at various time points for cFLIP and β-actin. Representative of 4 independent experiments. (B–C) WT and *CFLAR*⁻/⁻ THP-1 macrophages were pre-treated with inhibitors and then stimulated with LPS for 18–25 h. N = 3–4. (B) Cytotoxicity was measured by LDH release. (C) Caspase-3/7

activity was detected and quantified by Caspase-Glo 3/7. **(D–G)** *mCherry*- and *CFLAR*-overexpressing THP-1 macrophages were pre-treated with inhibitors and then stimulated with LPS or TNF or infected with *Ye* for 24 h. **(D)** Cytotoxicity was measured by LDH release. N = 5. **(E)** Cell viability was measured by ATP signal. N = 4. **(F, G)** Caspase-3/7 activity was detected and quantified by Caspase-Glo 3/7. Representative of 3 independent experiments performed in triplicate wells. ns, not significant, *p < 0.05, **p < 0.01, ***p < 0.001, ****p < 0.0001 by **(B,C)** Dunnett's multiple comparisons test, **(D,E)** Holm-Šídák method T test, or **(F,G)** Šídák's multiple comparisons test. All graphs depict mean + SD.

cells. RIPK1 was previously reported to contribute to apoptosis in both human Jurkat T cells and PANC-1 pancreatic tumor cells [17,73,74]. Given the lack of TLR4 expression in Jurkat cells [75], we stimulated the cells instead with TNF following IKK blockade, which largely phenocopies LPS+IKKi-induced apoptosis [12,76]. We then measured cell survival in the presence or absence of the RIPK1 inhibitor Nec-1. THP-1 macrophages and hMDMs underwent RIPK1 activity-independent cell death in response to TNF+IKKi treatment (**Fig 5A and 5B**), and hMDMs exhibited RIPK1 activity-independent caspase-8 cleavage (**S5A Fig**). These data demonstrate that in human macrophages, IKK blockade-induced cell death downstream of both LPS and TNF is RIPK1 activity-independent. Moreover, TNF+IKKi treatment in Jurkat cells (**Figs 5C and S5B**) and PANC-1 cells (**Figs 5D and S5C**) also induced caspase-8 cleavage and cell death that was not blocked by Nec-1, indicating that RIPK1 activity is dispensable for cell-extrinsic apoptosis in multiple human cell types in response to IKK blockade in the setting of TLR or TNF stimulation.

The E3 ubiquitin ligases cIAP1/2 ubiquitylate RIPK1 to form a scaffold for recruitment of signaling proteins, including IKKα/β and TAK1 [8,9,51,15]. SMAC mimetics (SM), which disrupt cIAPs and induce their degradation, are used extensively to induce RIPK1-dependent apoptosis. Interestingly, both THP-1 macrophages and hMDMs were resistant to cell death following TNF+SM treatment (**S5D and S5E Fig**), consistent with recent findings that non-polarized human macrophages are resistant to SM-induced cell death [77]. In contrast, both Jurkat T cells and PANC-1 cells were highly susceptible to caspase-8 cleavage and cell death following TNF+SM treatment, which were both reduced by Nec-1 treatment (**S5B, S5C, S5F and S5G Fig**). Collectively, these data indicate that distinct factors mediate cell-extrinsic apoptosis in different human cell types in response to IKK blockade or disruption of cIAPs.

## Discussion

Here, we report that unlike in murine macrophages, RIPK1 is dispensable for cell-extrinsic apoptosis in human macrophages during *Yersinia* infection or pharmacological blockade of IKK. Furthermore, overexpression of cFLIP partially protected human macrophages from IKK blockade-induced apoptosis, indicating that this cell death pathway is regulated by cFLIP downregulation. Additionally, we uncovered that in human Jurkat T cells and pancreatic epithelial cells, RIPK1 activity is also dispensable for IKK-blockade induced apoptosis. Overall, our findings illustrate that multiple human cell types undergo RIPK1 activity-independent apoptosis following IKK blockade.

Interestingly, contrary to murine macrophages, we found that human macrophages do not undergo cell death in response to *Yersinia pseudotuberculosis* (*Yptb*) infection. In contrast, *Y. enterocolitica* (*Ye*) induced robust cell death in human macrophages, which was dependent on YopP, a homolog of YopJ in *Yptb*. Notably, *Yptb* expressing YopP did not induce cell death (**S1C Fig**), indicating that other Yops in *Yptb* suppress death in human macrophages. Indeed, our recent studies indicate that *Yptb* effectors YopE/H/K synergistically suppress pyroptosis in both human IECs and macrophages, and that YopJ does not induce RIPK1-dependent apoptosis in human cells [78]. Why this does not occur with *Ye*, which in principle has highly overlapping effectors with *Yptb*, remains to be determined.

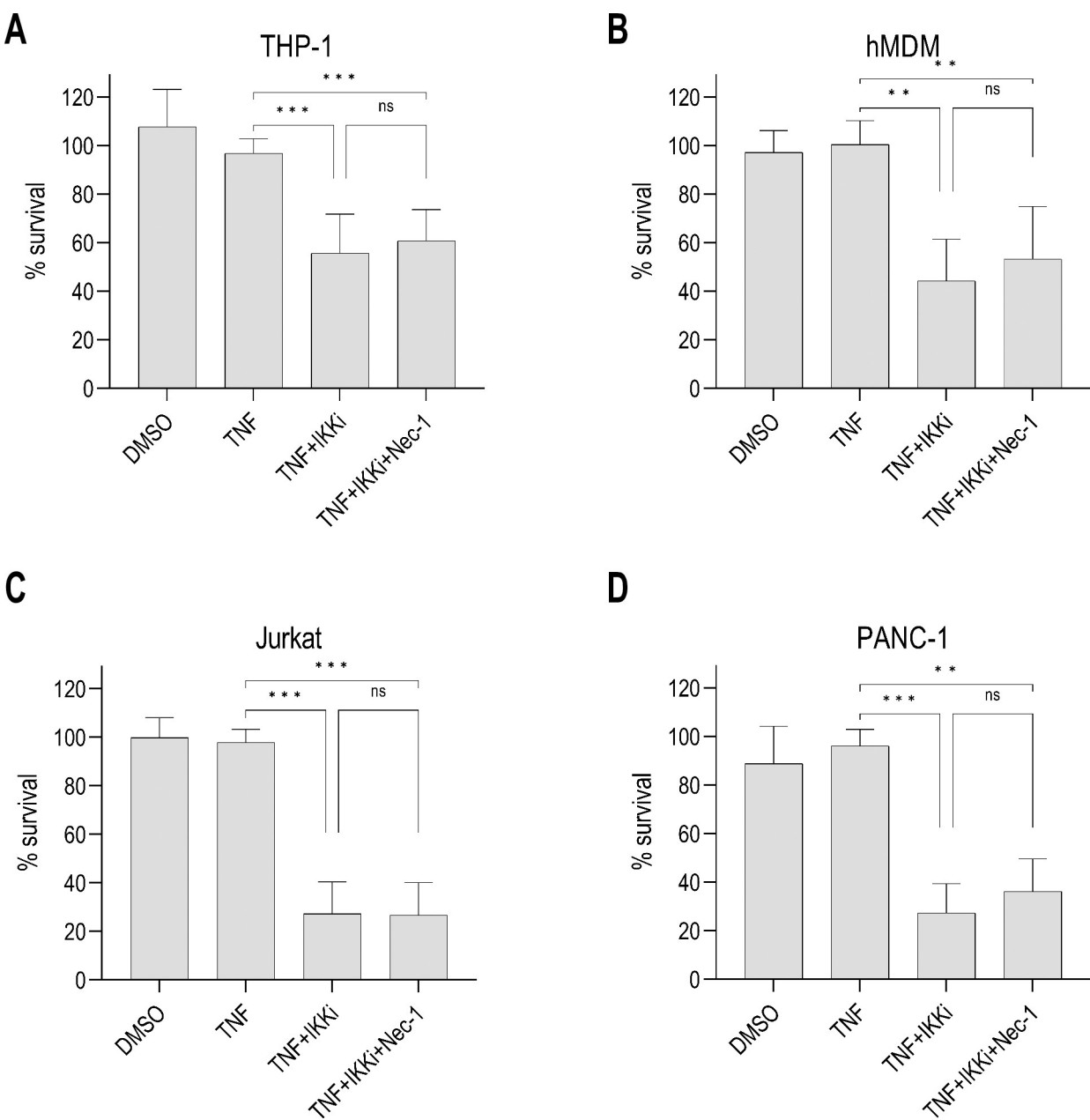

**Fig 5. Multiple human cell types undergo RIPK1 activity-independent cell death following IKK blockade.** Cells were pre-treated with inhibitors or vehicle control (DMSO) and then stimulated with TNF. Viability was measured by ATP levels in the following cell types: **(A)** THP-1 macrophages, 22–25 h, N = 6, **(B)** hMDMs, 5–6 h, N = 4, **(C)** Jurkat cells, 12–15 h, N = 3, **(D)** PANC-1 cells, 14–16 h, N = 3. ns, not significant, *p < 0.05, **p < 0.01, ***p < 0.001 by Tukey's multiple comparisons test. Graphs depict mean + SD.

Studies examining extrinsic apoptosis following Checkpoint 1 blockade in human cells have largely investigated SMAC mimetic treatment in epithelial cell lines and report that RIPK1 mediates apoptosis in this context [13,17,42,79], which we observed in Jurkat T cells and PANC-1 pancreatic epithelial cells as well (**S5 Fig**). Previous studies also established that RIPK1 is required for apoptosis following membrane-bound FasL treatment in human T cells [42,74,80]. In contrast, our findings demonstrate that RIPK1 is not required for cell-extrinsic apoptosis following IKK blockade in human macrophages, T cells, and pancreatic epithelial

cells, indicating that distinct regulatory mechanisms exist in human cells to mediate cell-extrinsic apoptosis.

Within its role as a scaffold within Complex I, RIPK1 mediates canonical NF-κB signaling and promotes host cell survival in numerous systems [40,42,64,81]. However, our data demonstrate that in the absence of RIPK1, human macrophages do not exhibit basal susceptibility to necroptosis, in contrast to previous reports [8,17,40,41,65,66,82], and these cells can still release substantial, albeit reduced, levels of TNF (Fig 3A), suggesting that RIPK1 is not entirely required for pro-inflammatory and pro-survival signaling. Furthermore, human macrophages did not undergo cell death in response to cIAP1/2 inhibition (S5D and S5E Fig). Given that cIAP1/2 contribute to pro-survival signaling via RIPK1, cIAP1/2 inhibition was not cytotoxic to human macrophages, and RIPK1$^{-/-}$ cells are not basally susceptible to necroptosis and can release TNF, it is possible that pro-survival signaling complexes in human macrophages are partially independent of cIAP1/2 and RIPK1. Overall, combined with our findings that IKK blockade induces RIPK1-independent cell death in human macrophages (Fig 3), our studies also suggest the possibility that a RIPK1-independent checkpoint promotes pro-survival NF-κB signaling and protects human macrophages from aberrant cell death. We cannot exclude the possibility that human macrophages can undergo RIPK1-dependent apoptosis in response to different stimuli. Nonetheless, our findings demonstrate that human and murine macrophages exhibit distinct requirements for RIPK1 in IKK blockade-induced apoptosis.

PANC-1 and Jurkat cells underwent apoptosis in response to treatment with SMAC mimetic and TNF, and this death was partially RIPK1-dependent (S5 Fig). In contrast, upon TNF+IKKi treatment, PANC-1 and Jurkat cells underwent RIPK1 activity-independent apoptosis (Fig 5). We speculate that in these cells, blocking cIAP1/2 destabilizes Complex I and releases RIPK1 to form Complex II and mediate apoptosis, whereas blocking IKK does not release RIPK1 but directly targets downstream NF-κB signaling. This is in contrast to murine cells, wherein blockade of TAK1 or IKK releases RIPK1 to mediate Complex II formation and apoptosis [15,34]. Overall, these findings reveal cell type- and species-specific differences in regulation of cell-extrinsic apoptosis in response to blockade of immune signaling.

The adaptor protein FADD is canonically understood to stabilize the caspase-8:cFLIP pro-survival complex as well as assemble the caspase-8:caspase-8 Complex II following cell-extrinsic apoptosis stimuli. In agreement with the field, we found that FADD protects human macrophages from necroptosis and is largely required for IKK blockade-induced apoptosis. In order to characterize the molecular regulation of this apoptosis pathway, further studies will be required to dissect the composition of the FADD-assembled cell death complex, both in the presence and absence of RIPK1.

RIPK1 is implicated in a broad range of human diseases [43,44,46], and there is significant interest in development of therapeutics to target RIPK1 kinase activity in a variety of human inflammatory diseases, neurodegenerative conditions, and cancer metastasis [46–48]. Extensive analyses in murine models highlight a key role for RIPK1 activity in promoting cell-extrinsic apoptosis and downstream responses. In contrast, our results indicate that there are key differences in the contribution and function of RIPK1 activity in human cells in a cell type- and stimulus-specific manner. Our findings highlight the need to further dissect the contributions of RIPK1 to cell death in different contexts in human cells.

## Materials and methods

### Ethics statement

All studies involving primary human macrophages were performed in compliance with the requirements of the US Department of Health and Human Services and the principles

expressed in the Declaration of Helsinki. Samples were obtained from the University of Pennsylvania Human Immunology Core. These samples are considered to be a secondary use of de-identified human specimens and are exempt via Title 55 Part 46, Subpart A of 46.101 (b) of the Code of Federal Regulations. All experiments performed with mouse bone marrow-derived macrophages were approved by the Institutional Animal Care and Use Committee of the University of Pennsylvania (protocol 804523).

## Cell cultures

All cells were maintained in a humidified incubator kept at 37°C with 5% $CO_2$. **Murine bone marrow-derived macrophages** (BMDMs) were isolated and differentiated as previously described [83] and replated in a 96-well tissue culture (TC)-treated plate at $0.5 \times 10^5$ cells/well. **Primary human monocyte-derived macrophages** (hMDMs) were differentiated as previously described [37] and replated in a 96-well TC-treated plate at $0.3 \times 10^5$ cells/well, or a 48-well tissue culture-treated plate at $1 \times 10^5$ cells/well. **THP-1 monocytes** (TIB-202; American Type Culture Collection) were maintained, differentiated, and replated as previously described [38] in a 96-well TC-treated plate at $0.5 \times 10^5$ cells/well, or a 48-well TC-treated plate at $2 \times 10^5$ cells/well. **Jurkat T cell clone E6-1** (TIB-152; American Type Culture Collection) was kindly provided by Will Bailis (University of Pennsylvania, Philadelphia). Cells were maintained in THP-1 media [38] as recommended. The day before the experiment, cells were replated in media without antibiotics in a 96-well TC-treated plate at $0.5 \times 10^5$ cells/well. **PANC-1** pancreatic epithelial carcinoma cells (CRL-1469; American Type Culture Collection) were maintained in DMEM supplemented with 10% (vol/vol) heat-inactivated fetal bovine serum (FBS), 100 IU/mL penicillin, and 100 μg/mL streptomycin. The day before the experiment, cells were detached with trypsin-EDTA (0.25%) and replated in media without antibiotics in a 96-well TC-treated plate at $0.5 \times 10^5$ cells/well.

## Generation of CRISPR/Cas9 THP-1 knockout cell lines

*RIPK1*[-/-], *CFLAR*[-/-], *FADD*[-/-], and *TRADD*[-/-] cells were generated using the CRISPR/Cas9 system as previously described [38]. Briefly, pLentiCRISPR v2 plasmids encoding the desired guide RNA (gRNA) and Cas9 were purchased from GenScript. The following target sequences were used (5' to 3'):

*RIPK1* gRNA 1: CGGCTTTCAGCACGTGCATC;
*CFLAR* gRNA 3: TGCCAATGCAATCGATTATC;
*FADD* gRNA 1: AGTCGTCGACGCGCCGCAGC;
*TRADD* gRNA 1: GGTGCGCGTAGGCATCCGAC.

Production of lentiviral particles, transduction of target cells, single-cell clone selection, and sequence validation were performed as previously described [38]. DNA was isolated from clones using DNeasy Blood & Tissue Kit (Qiagen, Hilden, Germany). The genomic region containing the target sequence was amplified by PCR using the following primers (all 5' to 3'): *RIPK1* Forward: CGTGGGAGTGATGTGTTGGA, *RIPK1* Reverse: ATCCTCCTGCCAAAAGTGCT, *CFLAR* Forward: ATGAACTTGTCTGGTTTGCAG, *CFLAR* Reverse: GCCTGCTTCCCTCTCTCTGTA. All *RIPK1*[-/-] and *CFLAR*[-/-] clones were sequence-validated and both alleles of each clone contained mutations which resulted in premature stop codons.

## Generation of *mCherry*- and *CFLAR*-overexpressing THP-1 cell lines

Human *CFLAR* was cloned into the pTwist Lenti SFFV Puro WPRE lentiviral vector backbone (Twist Bioscience). As a negative control, *mCherry* was cloned into pTwist Lenti SFFV Puro

**Table 1. Bacterial Strains.**

| Strain name | Reference/Source |
|---|---|
| IP2666 (WT *Yptb*) | From James Bliska [86,87] |
| IP26 (*ΔyopJ Yptb*) | From James Bliska [29] |
| IP26 pACYC184-YopP | [88] |
| JB580v (WT *Ye*) | From Virginia Miller [89] |
| YVM1612 (JB580v *ΔyopP Ye*) | From Virginia Miller |

WPRE vector with a C-terminal Streptag II (Twist Bioscience) [84]. Production of lentiviral particles, transduction, and selection of target cells were performed as previously described [38].

## Bacterial strains and growth conditions

*Yersinia* strains described in **Table 1** were grown and induced as previously described [85]. All cultures were washed and resuspended in pre-warmed serum-free media prior to infection. In all experiments, control cells were mock-infected with serum-free media.

## Stimulations and Infections

Cell media was replaced with fresh media containing inhibitors (**S1 Table**) and centrifuged at 300 x g for 1 min. 1–2 h following addition of inhibitors, cells were stimulated (**S1 Table**) or infected (**Table 1**). For priming conditions, cells were treated with 100 ng/mL *E. coli* LPS for 4 h before stimulation/infection. Cells were infected at a multiplicity of infection (MOI) of 20:1, unless otherwise indicated, centrifuged at 300 x g for 5 min, and incubated at 37°C. At 1 h post-infection, cells were treated with 100 µg/mL of gentamicin. At the indicated harvest time points, cells were centrifuged at 300 x g for 5 min prior to sample collection. All raw datasets are available on Dryad [90].

## siRNA-mediated knockdown

The Silencer Select siRNA oligo targeting human BID mRNA were purchased from Thermo Fisher Scientific (s1986, Catalog #4390824). The two Silencer Select negative control siRNAs (Silencer Select Negative Control No. 1 siRNA and Silencer Select Negative Control No. 2 siRNA) were purchased from Life Technologies (Ambion). The day before the transfection, THP-1 macrophages were differentiated and replated in a 48-well tissue culture-treated plate at 2–3×$10^5$ cells/well. siRNA-mediated knockdown was performed using Lipofectamine RNAiMAX Transfection Reagent (Thermo Fisher Scientific) and 7 pmol of total siRNA were transfected into cells according to the manufacturer's instructions. After four days of incubation, cells were harvested and/or treated for an experiment.

## LDH release

Cell supernatants were assayed for cytotoxicity by measuring loss of plasma membrane integrity via lactate dehydrogenase release, quantified using LDH Cytotoxicity Detection Kit (Sigma) according to the manufacturer's directions. % cytotoxicity was calculated by subtracting background (untreated/mock infected) and normalizing to maximal LDH release (1% Triton X-100).

## PI uptake

Propidium iodide (PI) (Thermo Fisher, Waltham, MA, USA) uptake was performed as previously described [91] and detected on a BioTek Synergy HT Multi-Detection Microplate Reader

(BioTek, Winooski, VT) at 485/20 excitation and 590/35 emission every 10 min for the indicated time points. % PI uptake was calculated by subtracting background (untreated) and normalizing to maximal PI uptake (1% Triton X-100).

## Cell viability

Cells were plated in a white-walled, clear-bottom TC-treated 96 well plate and treated as indicated. ATP was detected using the CellTiter-Glo 2.0 Assay Kit (Promega) according to the manufacturer's instructions, and the reaction was incubated in the dark for 30–60 min. Luminescence was read on a Gen5 plate reader (BioTek). % survival was calculated by subtracting background (1% Triton X-100) and normalizing to maximal ATP levels (untreated).

## Caspase-3/7 activity

Cells were plated in serum-free media in a white-walled, clear-bottom TC-treated 96 well plate and treated as indicated. Caspase-3/7 activity was detected using the Caspase-Glo 3/7 Assay Kit (Promega) according to the manufacturer's instructions. The reaction was incubated in the dark for 3 h. Luminescence was read on a Gen5 plate reader (BioTek) and values were normalized to cell signal read by ATP detection.

## Immunoblot analysis

Cells were lysed in SDS/PAGE sample buffer (50 mM Tris-HCl pH 6.8, 2% w/v SDS, 10% v/v glycerol, 0.1% Bromophenol Blue, 2 mM DTT). Lysates were boiled and centrifuged prior to running on 4–12% Bis-Tris gels (Thermo Fisher) and transferred to PVDF membranes. Membranes were immunoblotted using primary antibodies at 1:1000 dilution and HRP-linked secondary antibodies at 1:5000 dilution (**S1 Table**). Membranes were developed using Pierce ECL Plus and SuperSignal West Femto Maximum Sensitivity Substrate according to the manufacturer's instructions (Thermo Fisher).

## qRT-PCR

THP-1 cells were replated and differentiated as described above in a 48-well tissue culture-treated plate at a concentration of $2 \times 10^5$ cells/well. The next day, RNA was harvested (2 wells/condition) and isolated with RNeasy kit according to manufacturer instructions (Qiagen). cDNA synthesis was performed using High-Capacity cDNA Reverse Transcription Kit (Thermo Fisher). qPCR was performed using SYBR Green SuperMix (VWR International) on a QuantStudio Flex6000 (Thermo Fisher). The following primer sequences were used (all 5' to 3'):

*RIPK1* Forward: TTACATGGAAAAGGCGTGATACA
*RIPK1* Reverse: AGGTCTGCGATCTTAATGTGGA

## Cytokine release

Supernatants were harvested and cytokine levels were assayed using ELISA kits for human TNF and IL-18 (R&D Systems).

## Statistical analysis

Data were graphed and analyzed using GraphPad Prism 9 (San Diego, CA, USA) and presented as mean values + SD. Mean values were compared across conditions and P values were determined using one- or two-way analysis of variance (ANOVA) or t test, as indicated.

## Dryad DOI

10.5061/dryad.hmgqnk9rz.

## Supporting information

**S1 Fig. Characterization of *Yersinia*-induced cell death in human macrophages.** Cells were pre-treated with Nec-1 and then infected with the following strains of *Yersinia*: *WT Y. pseudo-tuberculosis (Yptb), ΔyopJ Yptb, ΔyopJ pYopP Yptb, WT Y. enterocolitica (Ye)*, or *ΔyopP Ye*. Cytotoxicity was measured by LDH release. **(A)** BMDMs were infected for 4–6 h at MOI 20. N = 3. **(B–C)** hMDMs were infected for 16–22 h. Each data point represents the mean of triplicate wells for each of 7–12 different human donor hMDMs. **(D)** Immunoblot analysis was performed on hMDM lysates for caspase-9 and β-actin. Representative of 2–3 independent experiments. ns, not significant, *p < 0.05, **p < 0.01, ***p < 0.001, ****p < 0.0001 by **(A)** Šídák's or **(B)** Tukey's multiple comparisons test. Graphs depict mean + SD. (TIF)

**S2 Fig. Characterization of cell death signaling downstream of IKK blockade in human macrophages.** Cells were pre-treated with inhibitors and then stimulated with LPS or infected with *Ye*. **(A)** hMDM cytotoxicity was measured by LDH release 5–7 h after stimulation, N = 3–6. **(B)** Immunoblot analysis was performed on hMDM lysates at various time points for phospho-RIPK1 (S166), RIPK1, and β-Actin. Representative of 2–4 independent experiments. **(C,D)** hMDM cytotoxicity was measured by LDH release after 10 h of stimulation. N = 3. THP-1 cell cytotoxicity was measured by LDH release 17–24 h after stimulation. N = 3–7. **(E)** Immunoblot analysis was performed on hMDM lysates for caspase-9 and β-Actin. Representative of 2–3 independent experiments. **(F–G)** THP-1 macrophages were transfected with siRNA specific for BID (siBid) or scrambled siRNA (siCTL) for 4 days, pre-treated with IKKi, and then stimulated with LPS. **(F)** Immunoblot analysis was performed on lysates 22 h after mock-treatment for BID and β-Actin. Representative of 4 independent experiments. **(G)** Cytotoxicity was measured by LDH release 22 h after stimulation. N = 4. **(H)** Depending on condition, THP-1 macrophages were LPS-primed (L_pr), pre-treated with IKKi, Nec-1, and/or MCC950, and then stimulated with LPS or Nigericin or infected with WT *Ye* for 23–24 h. IL-18 levels were measured by ELISA in the supernatant. N = 3. **(I–J)** Three independent *FADD⁻/⁻* THP-1 single-cell clonal cell lines were generated with CRISPR-Cas9. **(I)** Schematic representation of the *FADD* gene with exons (arrows), created using Benchling and BioRender. gRNA target sequence is highlighted in pink text. **(J)** Immunoblot analysis was performed on WT and *FADD⁻/⁻* THP-1 cell lysates for FADD and β-actin. **(K,L)** WT and *FADD⁻/⁻* THP-1 macrophages were pre-treated with inhibitors and then stimulated with LPS. Cytotoxicity was measured by LDH release 24–25 h post-stimulation. Representative of two independent experiments. **(M)** WT and *FADD⁻/⁻* THP-1 macrophages were infected with *Ye*. Caspase-3/7 activity was detected and quantified by Caspase-Glo 3/7 24–25 h post-infection. N = 3. ns, not significant, *p < 0.05, **p < 0.01, ***p < 0.001, ****p < 0.0001 by **(A,D,F)** Šídák's, **(C,G)** Tukey's, or **(K–M)** Dunnett's multiple comparisons test. Graphs depict mean + SD. (TIF)

**S3 Fig. Characterization of *RIPK1⁻/⁻* and *TRADD⁻/⁻* THP-1 macrophages.** Two independent *RIPK1⁻/⁻* single-cell clonal cell lines were generated with CRISPR-Cas9. **(A)** Schematic representation of the *RIPK1* gene with exons (arrows), created using Benchling and BioRender. gRNA target sequence is highlighted in pink text. **(B)** Sequence alignments of WT THP-1 and *RIPK1⁻/⁻* Clones 1 and 2 are shown for both alleles. Graphic was created using Benchling and BioRender. Red highlighting represents the mutated region. **(C)** Immunoblot analysis was

performed on WT and *RIPK1*$^{-/-}$ THP-1 cell lysates for RIPK1 and β-actin. **(D)** RT-qPCR was performed on WT and *RIPK1*$^{-/-}$ THP-1 cell lysates for RIPK1 expression relative to HPRT. **(E)** Immunoblot analysis was performed on WT and *RIPK1*$^{-/-}$ THP-1 cell lysates 5–6 h after stimulation or infection for caspase-3 and β-actin. Representative of 2–3 independent experiments. **(F)** Immunoblot analysis was performed on WT murine BMDM, hMDM, and THP-1 cell lysates for RIPK1, TRADD, and β-actin. **(G–H)** Bulk *TRADD*$^{-/-}$ THP-1 macrophages were generated with CRISPR-Cas9. **(G)** Schematic representation of the *TRADD* gene with exons (arrows), created using Benchling and BioRender. gRNA target sequence is highlighted in pink text. **(H)** Immunoblot analysis was performed on WT and bulk *TRADD*$^{-/-}$ THP-1 cell lysates for TRADD and β-actin.
(TIF)

**S4 Fig. Characterization of *CFLAR*$^{-/-}$ and plenti-*CFLAR* THP-1 macrophages. (A)** hMDMs were pre-treated with IKKi or Chx and then stimulated with LPS. Immunoblot analysis was performed on lysates at various time points for cFLIP and β-actin. Representative of 2 independent experiments**. (B–D)** 2 independent *CFLAR*$^{-/-}$ single-cell clonal cell lines were generated with CRISPR-Cas9. **(B)** Schematic representation of the *CFLAR* gene with exons (arrows), created using Benchling and BioRender. gRNA target sequence is highlighted in pink text. **(C)** Sequence alignments of WT THP-1 and *CFLAR*$^{-/-}$ Clones 1 and 2 are shown for both alleles. Red highlighting represents the mutated region. **(D)** Immunoblot analysis was performed on WT and *CFLAR*$^{-/-}$ THP-1 cell lysates for cFLIP and β-actin. **(E)** Immunoblot analysis was performed for cFLIP on lysates from plenti-*mCherry* and plenti-*CFLAR* stably-overexpressing THP-1 monocytes and PMA-differentiated macrophages. **(F)** plenti-*mCherry* and plenti-*CFLAR* stably-overexpressing THP-1 macrophages were pre-treated with IKKi and then stimulated with LPS. Immunoblot analysis was performed on lysates 5 h after stimulation for cleaved caspase-8, caspase-9, and β-actin, representative of 2–3 independent experiments.
(TIF)

**S5 Fig. Jurkat and PANC-1 cells undergo RIPK1-dependent cell death following cIAP1/2 blockade.** Cells were pre-treated with inhibitors or vehicle control (DMSO) and then stimulated with TNF. **(A–C)** Immunoblot analysis was performed on lysates 5–6 h after stimulation for cleaved caspase-8 and β-actin in the following cell types, representative of 2–3 independent experiments: **(A)** hMDMs, **(B)** Jurkat cells, and **(C)** PANC-1 cells. **(D–G)** Viability was measured by ATP signal in the following cell types: **(D)** THP-1 macrophages, 22–25 h, N = 4, **(E)** hMDMs, 5–6 h, N = 3, **(F)** Jurkat cells, 12–15 h, N = 4, and **(G)** PANC-1 cells, 14–16 h, N = 4. ns, not significant, *p $< 0.05$, **p $< 0.01$, ***p $< 0.001$, ****p $< 0.0001$ by Tukey's multiple comparisons test. Graphs depict mean + SD.
(TIF)

**S1 Table. Reagents.**
(XLSX)

## Acknowledgments

We thank Drs. Virginia Miller and Kimberly Walker for generously providing JB580v (WT *Y. enterocolitica*) [89] and YVM1612 (JB580v *ΔyopP*) for our studies. We thank members of the Shin and Brodsky labs for scientific discussion.

## Author Contributions

**Conceptualization:** Neha M. Nataraj, Sunny Shin, Igor E. Brodsky.

**Data curation:** Neha M. Nataraj, Reyna Garcia Sillas.

**Formal analysis:** Neha M. Nataraj, Sunny Shin, Igor E. Brodsky.

**Funding acquisition:** Neha M. Nataraj, Sunny Shin, Igor E. Brodsky.

**Investigation:** Neha M. Nataraj, Reyna Garcia Sillas, Beatrice I. Herrmann, Sunny Shin, Igor E. Brodsky.

**Methodology:** Neha M. Nataraj.

**Project administration:** Neha M. Nataraj.

**Supervision:** Sunny Shin, Igor E. Brodsky.

**Writing – original draft:** Neha M. Nataraj, Sunny Shin, Igor E. Brodsky.

**Writing – review & editing:** Neha M. Nataraj, Sunny Shin, Igor E. Brodsky.

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
