## [Decision Letter · Decision Letter 0]

18 Sep 2023

Dear Drs Brodsky and Shin,

Thank you very much for submitting your manuscript "Blockade of IKK signaling induces RIPK1-independent apoptosis in human cells" for consideration at PLOS Pathogens.

Your manuscript was reviewed by three independent experts in your field of research. They are all enthusiastic about your work. All agree that the question raised by your work is very important, timely, and of general interest. However, all three raise major questions (conceptual, methodological and mechanistic). One reviewer requests to perform some experiments at early time points than the one used in your study to support your conclusions. From a mechanistic point of view, the importance of TRADD (between species), the contribution of caspase 9, and the possibility that caspase 8 is activated by a protein complex other than the one proposed are also raised by the reviewers. One reviewer also highlights the benefits of using RIPK1-deficient cells for your study. Overall, the reviewers provided key experiments to support your conclusions and reinforce the mechanistic aspect of your study. Below you'll find the three reviews with detailed concerns.

We cannot make any decision about publication until we have seen the revised manuscript and your response to the reviewers' comments. Your revised manuscript will be sent to reviewers for further evaluation.

Sincerely,

Florent Sebbane

Guest Editor

PLOS Pathogens

Karla Satchell

Section Editor

PLOS Pathogens

Kasturi Haldar

Editor-in-Chief

PLOS Pathogens

orcid.org/0000-0001-5065-158X

Michael Malim

Editor-in-Chief

PLOS Pathogens

orcid.org/0000-0002-7699-2064

Dear Drs Brodsky and Shin,

Three experts in your field of research have reviewed your manuscript. They are all enthusiastic about your work. All agree that the question raised by your work is very important, timely, and of general interest. However, all three raise major questions (conceptual, methodological and mechanistic). One reviewer request to perform some experiments at early time points than the one used in your study to support your conclusions. From a mechanistic point of view, the importance of TRADD (between species), the contribution of caspase 9, and the possibility that caspase 8 is activated by a protein complex other than the one proposed are also raised by the reviewers. One reviewer also highlights the benefits of using RIPK1-deficient cells for your study. Overall, the reviewers provided key experiments to support your conclusions and reinforce the mechanistic aspect of your study. Below you'll find the three reviews with detailed concerns.

Please return your revised manuscript within 2 months; if you need more time to respond to the reviewers’ comments, please contact us.

Thank you for submitting your manuscript to PLoS Pathogens

Sincerely,

Florent Sebbane

Reviewer's Responses to Questions

**Part I - Summary**

Reviewer #1: In the manuscript by Nataraj et al, the authors discovered that in human cells RIPK1 is not required for the induction of apoptosis triggered by IKK inhibition, either pharmacologic or by the Yersinia virulence factor YopJ. Indeed, they show that, unlike in mouse cells, in human cells, including human monocyte-derived macrophages and THP-1, Yersinia infection or LPS treatment combined with IKK pharmacologic inhibition induce RIPK1-independent apoptosis. Of note, neither the kinase nor the scaffolding activity of RIPK1 is required for apoptosis induction in these settings. Instead, IKK inhibition-mediated cell death could be partially rescue by cFLIP overexpression, consistently with the fact that IKK inhibition leads to cFLIP downregulation.

Over the past decade, mouse studies have highlighted the role of RIPK1 kinase activity in a number of cell death-driven inflammatory pathologies, including Yersinia infection, Sharpin mutant-driven dermatitis and TNF-driven lethal sepsis. However, it is still not clear the contribution of RIPK1 kinase activity to inflammatory syndromes in human and whether RIPK1 kinase inhibition represents a valid therapeutic approach to treat inflammation-related diseases. Therefore, there is the need to elucidate the role of RIPK1 kinase activity in the execution of cell death downstream of immune receptors such as TNFR1 and TLRs in human cells. Hence, the notion that RIPK1 is not involved in apoptosis upon IKK blockade is of general interest for the entire cell death and inflammation field. In addition, the evidence provided in this study suggest that RIPK1 inhibition would not interfere with the ability of the host to clear Yersinia.

Overall, the experiments are well performed, with the most appropriate controls. The results are clearly presented and conclusions justified. However, some additional experiments would be required to strengthen the mechanistic aspect of the study.

Reviewer #2: The study of Neha Nataraj and colleagues concerns a potentially important issue. By applying Yersinia enterocolitica as tool for mediating infection-related cell death the role of IKK in controlling the cytotoxic activity of RIPK1 was investigated in human cells. Yersinia is well characterized to induce apoptosis in murine macrophages and the exploration of the mechanisms involved had previously provided valuable new insights into the regulation of cellular life and death during infection. However, it is less clear whether the findings obtained on murine cells correspond well with human cell infection. Surprisingly, the current work now indicates that RIPK1, which has been identified as critical control element in the cell death modalities of mouse cells, is dispensable for Yersinia-induced death of human macrophages. It is thus suggested that RIPK1 may fulfil distinct tasks in human cell biology which could potentially be relevant for therapeutic targeting of RIPK1 in human diseases.

The study addresses a timely and interesting question. There are, however, some conceptual and methodological concerns that should be specified to substantiate the conclusions of the study.

Reviewer #3: The manuscript by N. Nataraj et al. addresses species-specific differences in host response to Yersinia infection, particularly effector-mediated arrest imposed by bacteria on the inflammatory response in macrophages. Key differences in host response between humans and mice need to be explored, which is a very interesting question thus making significant the presented studies. The rationale is provided by their own observations as well as from reports by other groups. Instrumental in this paper Y. enterocolitica species of bacteria that disrupts inflammatory signaling via inhibition of IKK-complex. In summary, my impression of the novelty and significance is very positive.

**Part II – Major Issues: Key Experiments Required for Acceptance**

Reviewer #1: 1) The authors clearly showed that in hMDM Yersinia- and LPS/IKKi-induced apoptosis is RIPK1 independent and Caspase-8-dependent. Accordingly, they observed cleaved Caspase-3. In absence of RIPK1, the only known DD-containing protein that can nucleate FADD and Caspase-8 for this last to get activated is TRADD. The authors should compare RIPK1 and TRADD expression between mBMDMs and hMDM to role in or out the possibility that the difference between human and mouse is due to the different expression levels of RIPK1 and TRADD. Once can indeed envisage the scenario where in human macrophages, unlike mouse cells, TRADD is significantly more abundant than RIPK1 and therefore it is the main molecule responsible for Caspase-8 activation.

Furthermore, the authors should address whether TRADD deletion or downregulation blocks/attenuate IKK-inhibition mediated death.

2) Yersinia infection and LPS/IKKi treatments also induce Caspase-9 activation. The authors should determine the contribution Caspase-9 activation to the apoptotic phenotype, for example by deleting/downregulating Bid or by over-expressing Bcl-2.

3) In figure 4 it would be useful to have the cleaved-Caspase-8 and cleaved-Caspase-9 immunoblotting for the cFLIP overexpressing cells infected with Yersinia or treated with LPS/IKKi. This would indicate whether the rescue of the apoptotic phenotype is partial only because of incomplete inhibition of Caspase-8 by cFLIP.

4) In figure 5, the cleaved-Caspase-8 immunoblotting should be added for all the 4 cell lines.

Reviewer #2: 1. The time courses of the investigated events require more precise synchronization. It seems that Yersinia- / IKK inhibition-induced cell death starts already at around 5 to 7 h (Fig. 1C, 2A, 2B, S1D). However, processing of caspase-8 and caspase-3 was investigated at much later time points (18 h). As caspase-8 is supposed to be the upstream initiator caspase it should be analysed whether capase-8 activation precedes onset of apoptosis and processing of caspase-9 (5 h, Fig S1D).

2. Similarly, the role of RIPK1 investigated in Fig. 3 should be evaluated also in the early phase of cell death initiation. At the late time points in Fig. 3 the cells may already have switched to RIPK1-independent cell death modes and a potential involvement of RIPK1 in cell death initiation may have been missed.

3. In the same line, the phosphorylation of RIPK1 should be assessed at early time points, as the RIPK1 kinase activity and phosphorylation is supposed to be relevant for the initiation of cell death. It does not become clear from the manuscript at which time point RIPK1 phosphorylation was analysed in Fig. 2D. Fig. 2D and 2E are not correctly assigned in the figure caption.

Reviewer #3: I wish there were more data in characterization of RIPK1-deficient THP1 cell line, which is my major concern. In short, I would like to see figure 2 to be done in RIPK1-deficient cells. Another major concern is lack of any characterization (or attempt to characterize) of what would be the alternative complex responsible for activation of caspase 8. To start with RIPK1-KO cells, caspase 8 is inactive in unstimulated cells. If it is not complexed with RIPK1, it must be bound to cFLIP? Does FADD get involved with caspase 8 in stimulated cells? For instance, caspase 8 could be pulled down and examined on the presence of FADD or cFLIP. Similarly, in kinase inhibited conditions (Figure 2), does RIPK1 bind caspase 8? As one of the ways to improve the manuscript, I would recommend confining it to the knockout instead of the kinase inactive conditions, which would release some space thus making the story more focused.

**Part III – Minor Issues: Editorial and Data Presentation Modifications**

Reviewer #1: N/A

Reviewer #2: 4. As the study indicates that RIPK1 is dispensable for Yersinia- / IKK inhibition-induced apoptosis in human macrophages it could be worth to address experimentally whether the cell death-related, so called “ripoptosome”, known from mouse cell death models, forms in human cells in the investigated conditions.

5. The results shown in Fig. 4 suggest that downregulation of cFLIP-L expression is a major event responsible for Yersinia- / IKK inhibition-mediated apoptosis. To substantiate a possible coherence between protein synthesis blockade and initiation of cell death following Yersinia / LPS / TNF stimulation the data obtained with the cFLIP-L-knockout and -overexpressing cells could be complemented by experiments using cycloheximide instead of IKKi as protein synthesis inhibitor. Fig. 4C, which displays caspase activities, has an incorrect figure caption.

6. It is concluded from Fig. 5 that IKK blockade alone does not release the cytotoxic activity of RIPK1 in human cells. As it is known that other kinases besides IKK may prevent RIPK1 from signalling cell death it could be useful to analyse a TAK1 inhibitor in the experiments of Fig. 5. TAK1 acts upstream of IKK and controls additional survival-promoting signals that may be more important for controlling RIPK1 in human cells than in mouse cells. The inhibition of TAK1 would further more closely resemble the cellular YopP/YopJ effects than IKK blockade alone.

7. Is there an explanation why a high MOI may apparently impede Yersinia in triggering cell death (Fig. S1B)?

8. The title suggests that blockade of IKK signalling in general mediates RIPK1-independent apoptosis in human cells. However, this has been formally addressed only for THP-1 cells by using Nec-1 and RIPK1-knockout cells. The experiments on the other human cell types solely rely on Nec-1 application which does not rule out that presence of RIPK1 by itself may be required for cell death induction independently from its kinase activity.

Reviewer #3: there are several minor issues left to be addressed after the major issues are addressed.

PLOS authors have the option to publish the peer review history of their article (what does this mean?). If published, this will include your full peer review and any attached files.

Reviewer #1: No

Reviewer #2: No

Reviewer #3: No
---

## [Decision Letter · Decision Letter 1]

3 Jul 2024

Dear Drs Brodsky and Shin,

As you will read below, the three reviewers who examined the first version of your manuscript are very enthusiastic about your revised manuscript. You have addressed all their concerns. However, you will need to address two new minor points raised by one of the reviewers. Therefore, we are likely to accept your manuscript for publication, providing that you modify the manuscript according to the review recommendations.

Sincerely,

Florent Sebbane

Guest Editor

PLOS Pathogens

Florent Sebbane

Guest Editor

PLOS Pathogens

Michael Malim

Editor-in-Chief

PLOS Pathogens

orcid.org/0000-0002-7699-2064

Dear Drs Brodsky and Shin,

As you will read below, the three reviewers who examined the first version of your manuscript are very enthusiastic about your revised manuscript. You have addressed all their concerns. However, you will need to address two new minor points raised by one of the reviewers.

Sincerely,

Florent Sebbane

Reviewer Comments (if any, and for reference):

Reviewer's Responses to Questions

**Part I - Summary**

Reviewer #1: Over the past decade, mouse studies have highlighted the role of RIPK1 kinase activity in a number of cell death-driven inflammatory pathologies, including Yersinia infection, Sharpin mutant-driven dermatitis and TNF-driven lethal sepsis. However, it is still not clear the contribution of RIPK1 kinase activity to inflammatory syndromes in human and whether RIPK1 kinase inhibition represents a valid therapeutic approach to treat inflammation-related diseases. Therefore, there is the need to elucidate the role of RIPK1 kinase activity in the execution of cell death downstream of immune receptors such as TNFR1 and TLRs in human cells. Hence, the notion that RIPK1 is not involved in apoptosis upon IKK blockade is of general interest for the entire cell death and inflammation field. In addition, the evidence provided in this study suggest that RIPK1 inhibition would not interfere with the ability of the host to clear Yersinia.

Reviewer #2: Neha Nataraj and colleagues resubmitted a revised manuscript of their study on the role of RIPK1 in mediating human macrophage cell death following Yersinia infection. The authors have addressed the comments previously raised by the reviewers and consequently provided additional data and information. This increased the plausibility and significance of the work. I thus feel that all major and relevant issues were convincingly resolved.

Reviewer #3: This is a revised submission of the manuscript by by Dr. Igor E Brodsky and colleagues, which fully addressed all the concerns raised in the initial submission. I have no further question and enthusiastically recommend its acceptance

**Part II – Major Issues: Key Experiments Required for Acceptance**

Reviewer #1: The authors addressed experimentally all my concerns and points of criticism and significantly improved the quality of the study compared to the first submission.

Reviewer #2: (No Response)

Reviewer #3: (No Response)

**Part III – Minor Issues: Editorial and Data Presentation Modifications**

Reviewer #1: (No Response)

Reviewer #2: There are only two minor points originating from the revision that may require additional attention:

1. Fig. S2B indicates that RIPK1 is rapidly processed and cleaved by caspases following LPS and IKKi treatment. Did the authors consider that the independency of the described death of human macrophages from RIPK1 may result from rapid RIPK1 degradation, consequently entailing RIPK1-independent cell death?

2. Are the authors sure that the TRADD antibody applied in Fig. S3F is cross-species-reactive to mouse and human as proposed in the comment to point #1 of reviewer one? The antibody listed in Table 2 seems to be produced from human TRADD and specificity is stated just for human TRADD, according to the information provided by the manufacturer.

Reviewer #3: (No Response)

PLOS authors have the option to publish the peer review history of their article (what does this mean?). If published, this will include your full peer review and any attached files.

Reviewer #1: No

Reviewer #2: No

Reviewer #3: No

Figure Files:

Data Requirements:

Reproducibility:

References:

---

## [Editor Report · Decision Letter 2]

31 Jul 2024

Dear Dr. Brodsky,

We are pleased to inform you that your manuscript 'Blockade of IKK signaling induces RIPK1-independent apoptosis in human macrophages' has been provisionally accepted for publication in PLOS Pathogens.

Best regards,

Florent Sebbane

Guest Editor

PLOS Pathogens

Florent Sebbane

Guest Editor

PLOS Pathogens

Michael Malim

Editor-in-Chief

PLOS Pathogens

orcid.org/0000-0002-7699-2064
---

## [Editor Report · Acceptance letter]

21 Aug 2024

Dear Dr. Brodsky,

We are delighted to inform you that your manuscript, "Blockade of IKK signaling induces RIPK1-independent apoptosis in human macrophages," has been formally accepted for publication in PLOS Pathogens.

Best regards,

Michael Malim

Editor-in-Chief

PLOS Pathogens

orcid.org/0000-0002-7699-2064